# Taxonomy, Ecology, and Cellulolytic Properties of the Genus *Bacillus* and Related Genera

Jakub Dobrzyński [1,*], Barbara Wróbel [1] and Ewa Beata Górska [2]

[1] Institute of Technology and Life Sciences—National Research Institute, Falenty, 3 Hrabska Avenue, 05-090 Raszyn, Poland; b.wrobel@itp.edu.pl
[2] Department of Biochemistry and Microbiology, Institute of Biology, Warsaw University of Life Sciences—SGGW, 02-776 Warsaw, Poland; e.b.gorska@wp.pl
[*] Correspondence: j.dobrzynski@itp.edu.pl

**Abstract:** Bacteria of the genus *Bacillus* and related genera (e.g., *Paenibacillus*, *Alicyclobacillus* or *Brevibacillus*) belong to the phylum Firmicutes. Taxonomically, it is a diverse group of bacteria that, to date, has not been well described phylogenetically. The group consists of aerobic and relatively anaerobic bacteria, capable of spore-forming. *Bacillus* spp. and related genera are widely distributed in the environment, with a particular role in soil. Their abundance in the agricultural environment depends mainly on fertilization, but can also depend on soil cultivated methods, meaning whether the plants are grown in monoculture or rotation systems. The highest abundance of the phylum Firmicutes is usually recorded in soil fertilized with manure. Due to the great abundance of cellulose in the environment, one of the most important physiological groups among these spore-forming bacteria are cellulolytic bacteria. Three key cellulases produced by *Bacillus* spp. and related genera are required for complete cellulose degradation and include endoglucanases, exoglucanases, and β-glucosidases. Due to probable independent evolution, cellulases are encoded by hundreds of genes, which results in a large structural diversity of these enzymes. The microbial degradation of cellulose depends on its type and environmental conditions such as pH, temperature, and various substances including metal ions. In addition, *Bacillus* spp. are among a few bacteria capable of producing multi-enzymatic protein complexes called cellulosomes. In conclusion, the taxonomy of *Bacillus* spp. and related bacteria needs to be reorganized based on, among other things, additional genetic markers. Also, the ecology of soil bacteria of the genus *Bacillus* requires additions, especially in the identification of physical and chemical parameters affecting the occurrence of the group of bacteria. Finally, it is worth adding that despite many spore-forming strains well-studied for cellulolytic activity, still few are used in industry, for instance for biodegradation or bioconversion of lignocellulosic waste into biogas or biofuel. Therefore, research aimed at optimizing the cellulolytic properties of spore-forming bacteria is needed for more efficient commercialization.

**Keywords:** taxonomy; Firmicutes; spore-forming bacteria; cellulases





## 1. Introduction

*Bacillus* spp. and related genera, including *Paenibacillus*, *Alicyclobacillus*, or *Brevibacillus*, are mostly Gram-positive and have the ability to produce spores and display metabolic capabilities under aerobic as well as relatively anaerobic conditions (Figure 1). Due to their characteristics, bacteria of this group have high resistance to environmental stresses such as drought, water stress, UV radiation, or low nutrient content in the environment [1,2]. *Bacillus* spp. and related genera commonly populate the Earth and occur in a variety of environments of both natural and anthropogenic origin [2–4].

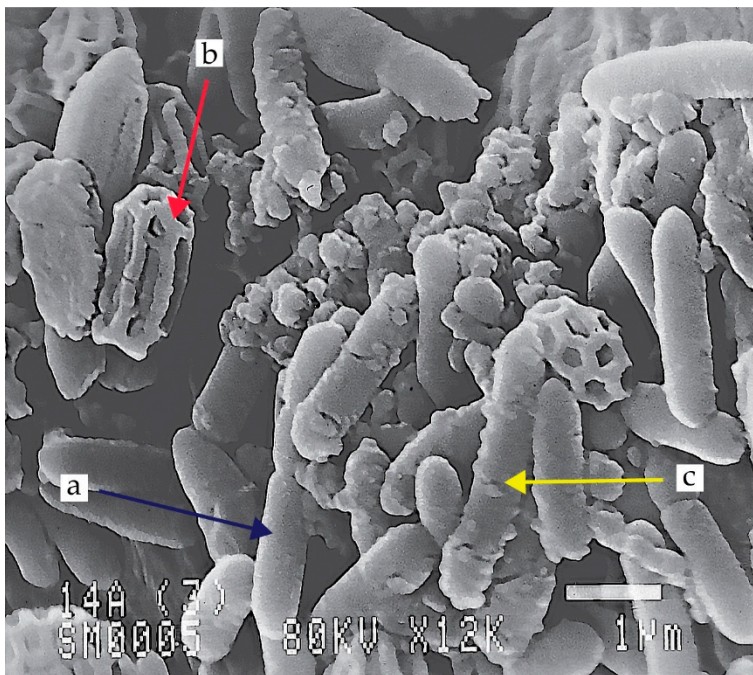

**Figure 1.** Scanning electron microscope (SEM) image of *Paenibacillus polymyxa* EG14 cultivated on medium with 0.5% cellobiose: vegetative cell (**a**), endospores (**b**), cellulosomes (**c**) (own photo).

In recent decades, rapid population growth has resulted in a significant intensification of agriculture, which has contributed to environmental pollution affecting both community structure and physiology of most microbial groups in the soil [5–7]. Because of the high cellulose abundance, organisms that have cellulolytic activity gained importance. Cellulases are synthesized by almost all groups of systematic organisms including microorganisms such as bacteria, fungi, protists, plants, and nematodes [8]. The bacteria capable of producing cellulolytic enzymes include both aerobic bacteria, e.g., *Butyrivibrio* spp. and *Cellulomonas* spp., as well as anaerobic bacteria, e.g., *Clostridium* spp. or *Ruminococcus* spp. bacteria [9]. However, due to their resistance to unfavorable environmental conditions, aerobic and relatively anaerobic, spore-forming bacteria of the phylum Firmicutes (i.e., bacterial strains of the genus *Bacillus* and related genera) are the most interesting [10]. So far, hundreds of cellulolytic spore-forming strains belonging to the phylum Firmicutes were isolated, including the genus *Bacillus* (e.g., *B. subtilis*) [11]; the genus *Alicyclobacillus* (e.g., *A. cellulosilyticus* [12] and *A. acidocaldarius* [13]); the genus *Geobacillus* (e.g., *Geobacillus* sp. HTA426) [14], or the genus *Lysinibacillus* (e.g., *Lysinibacillus fusiformis*) [15]. Despite the large number of isolated strains, still only a small part of them is commercialized, e.g., in the biodegradation of lignocellulosic waste.

The aim of the review is to summarize current knowledge of the taxonomy, ecology and properties of cellulolytic bacteria and to find gaps, the filling of which may lead to a better understanding of the ecology of *Bacillus* spp. and related genera, improving taxonomy and to a better exploitation of the cellulolytic potential of the bacteria group.

## 2. Taxonomy of the Genus *Bacillus* and Related Genera

The genus *Bacillus* and related genera (e.g., *Paenibacillus* or *Alicylobacillus*) are a very diverse group of bacteria that belongs to the phylum Firmicutes. The phylum includes several classes such as Bacilli, Clostridia, Mollicutes, and Erysipelotrichia. The group of bacteria which is the subject of the review belongs to the class Bacilli. Currently, it is classified into several families including Bacillaceae, Paenibacillaceae, etc. [16,17]. However, earlier, there was only the genus *Bacillus* which was first described in 1874. One of the first species specified in the genus *Bacillus* is type species-*B. subtilis*. The species is also one of the best-studied organisms belonging to the prokaryota and thus is extensively used as a model microor-

ganism for Gram-positive bacteria. Also, in the past, *B. subtilis* was the model organism in studies conducted to understand spore formation mechanisms [18,19]. However, despite numerous and extensive studies on *Bacillus* and related bacteria, the overall phylogenetic and evolutionary history of these genera remains unclear and relatively unexplored.

Initially, bacteria were identified by phenotypic methods using light microscopy and staining techniques including Gram staining [20]. Other older techniques that remain helpful nowadays include evaluation of bacterial biochemical properties, for example determining the metabolic profile, which can be used to differentiate between bacterial species. An example of such identification methods is the API® 50 CHB/E system, which is based on 50 biochemical tests that test the carbohydrate metabolism of the data from *Bacillus* spp. and related genera [21]. On the other hand, an improved version for bacterial identification using rapid tests is the Biolog OmniLog System. In addition to carbon source metabolism, the method also includes 23 chemical tests that determine, for example, the bacteria's tolerance to salinity or sensitivity to other chemicals [22]. In both cases, the obtained results can be compared with databases and, to some extent, determine the taxonomic affiliation of the studied bacteria. However, it was not until the development of sequencing techniques in the 1990s that major changes in the taxonomy of spore-forming bacteria occurred. Then, other genera began to be separated from the genus *Bacillus*. Most phylogenetic studies are based on 16S rRNA gene sequences [23]. Based on branching in phylogenetic trees, initial phylogenetic studies delineated and identified five clusters of *Bacillus* species [24]. One of these clusters including *B. subtilis* was named *Bacillus sensu stricto* [24], while bacterial species from the other clusters were subsequently reclassified to form the following genera: *Paenibacillus*, *Lysinibacillus Brevibacillus*, and *Geobacillus* [23,25,26]. In subsequent years, based on phylogenetic and phenotypic results, many other *Bacillus* species were reclassified to form several new genera, for instance *Aneurinibacillus*, *Alicyclobacillus*, *Alkalicoccus*, *Sporosarcina*, *Gracilibacillus*, *Virgibacillus*, *Hydrogenibacillus*, *Ureibacillus*, *Solibacillus*. [23,25,27,28]. The bacterial genera listed above belong to different families, as shown in Table 1. For instance, *Alicylobacillus*, along with *Tumebacillus*, *Effusibacillus*, *Kyrpidia*, and *Sulfobacillus*, have been assigned to the family Alicylobacillace; a particularly important genus among those listed is *Alicylobacillus* [29,30]. On the other hand, the Paenibacillace family includes 14 genera of spore-forming bacteria of which *Paenibacillus* and *Brevibacillus* are the most interesting in terms of potential industrial use [31].

**Table 1.** List of families assigned to the order Bacillales [17,23,29,32–39].

| Family Name | Proposed by | Type Genus | Other Example of Genus |
|---|---|---|---|
| Alicyclobacillaceae | da Costa and Rainey | *Alicyclobacillus* | *Effusibacillus, Kyrpidia, Tumebacillus* |
| Bacillaceae | Fischer | *Bacillus* | *Perribacillus, Weizmannia, Neobacillus, Metabacillus, Ferdinandcohnia, Gottfriedia, Heyndrickxia, Lederbergia* |
| Caryophanaceae | Peshkoff | *Caryophanon* | *Bhargavaea, Chryseomicrobium, Chungangia, Filibacter, Indiicoccus, Jeotgalibacillus, Kurthia, Lysinibacillus, Marinibacillus, Metalysinibacillus, Metaplanococcus, Metasolibacillus, Paenisporosarcina, Planococcus, Psychrobacillus, Rummeliibacillus, Savagea, Solibacillus, Sporosarcina, Ureibacillus* |
| Desulfuribacillaceae | Sorokin et al. | *Desulfuribacillus* | — |
| Listeriaceae | Ludwig et al. | *Listeria* | *Brochothrix* |

**Table 1.** *Cont.*

| Family Name | Proposed by | Type Genus | Other Example of Genus |
|---|---|---|---|
| Paenibacillaceae | De Vos et al. | *Paenibacillus* | *Ammoniibacillus, Aneurinibacillus group, (Ammoniphilus, Aneurinibacillus, Oxalophagus), Brevibacillus, Chengkuizengella, Cohnella, Fontibacillus, Gorillibacterium, Longirhabdus, Marinicrinis, Paludirhabdus, Saccharibacillus, Thermobacillus, Xylanibacillus* |
| Pasteuriaceae | Laurent | *Pasteuria* | — |
| Sporolactobacillaceae | Ludwig et al. | *Sporolactobacillus* | *Caenibacillus, Camelliibacillus Pullulanibacillus, Scopulibacillus, Sinobaca, Tuberibacillus* |
| Staphylococcaceae | Schleifer and Bell | *Staphylococcus* | *Abyssicoccus, Aliicoccus, Auricoccus Corticicoccus, Gemella, Jeotgalicoccus Macrococcus, Nosocomiicoccus, Salinicoccus* |
| Thermoactinomycetaceae | Matsuo et al. | *Thermoactinomyces* | *Baia, Croceifilum, Desmospora, Geothermomicrobium, Hazenella, Kroppenstedtia, Laceyella, Lihuaxuella, Marininema, Marinithermofilum, Mechercharimyces, Melghirimyces, Novibacillus, Paludifilum, Planifilum, Polycladomyces, Risungbinella, Salinithrix, Seinonella, Shimazuella, Thermoflavimicrobium* |

Although studies using sequences coding 16S rRNA have led to the reclassification of many species to new genera, according to many researchers, analyzes based on this variable gene are not fully sufficient to correctly distinguish taxa at the species level [40–43]. Similarly to other taxa, the previous classification of order Bacillales and other related orders was mostly based on 16S rRNA gene sequences. Moreover, research based on this type of analysis contributes to the formation of various types of anomalies. The occurrence of anomalies among the order is confirmed by the fact that several families and genera forming spores and non-spores were placed in it. Such patterns suggest that one gene marker is not sufficient to determine the phylogenetic structure of the Bacillales order [41]. Phylogenetic analyses have also been carried out using several other gene or protein sequences [44–46]. However, due to the relatively small number of *Bacillus* species studied in these researches, the analysis is insufficient to elucidate species relationships within this large genus. Consequently, *Bacillus* spp. is still a highly heterogeneous genus characterized by extensive polyphyletic branching with other genera of the family Bacillaceae [47,48]. Furthermore, as a result of the diverse branching of current species in the genus *Bacillus*, it was difficult to limit the addition of new species to this genus, even despite the large differences between the new species and the type species. Therefore, more valid methods should be studied and used to delineate the genus *Bacillus* and limit the placement of unrelated species within it [23]. For instance, comparative analysis of whole genomes (based on NCBI available sequences/genomes) makes it possible to study the evolutionary relations of species, and thus provide opportunities to identify molecular markers (molecular synpomorphies) [23,49]. For example, molecular synapomorphies that contain conserved insertions and signature deletions in protein sequences are good means of differentiating species from the two major clades of the genus *Bacillus*, i.e., the "Subtilis clade" and the "Cereus clade". According to ICNP rule 56a, the transfer of a species from the Cereus clade to a new genus may play some part in human health; therefore, transfer to another species is not advisable. As evidenced by a comprehensive genomic analysis of Bacillaceae species, 36 new genetic markers (i.e., conserved signature indels (CSIs)) were detected [23]. Importantly, based on new CSIs, the monophyletic groups found in all reconstructed or new phylogenetic trees were named as follows: Simplex, Firmus, Alcalophilus, Niacini, Fastidiosus, and Jeotgali clades, and collectively included



from 5 to 23 *Bacillus* species. In addition, researchers also performed a phylogenomic analysis on various Firmicutes proteins including core and conserved proteins. Moreover, the combined sequences of highly conserved proteins such as GyrB, GyrA, RpoC, RpoB, UvrD, or PolA were also studied, and confirmed by an extended comparative analysis of the genome of the above-mentioned protein sequences [23]. The authors of this study, based on robust evidence from many lines of research (conducted in parallel) confirming the existence of six distinct *Bacillus* clades, propose the transfer of species from these clades to six novel genera of Bacillaceae family, namely *Alkalihalobacillus* gen. nov., *Cytobacillus* gen. nov., *Mesobacillus* gen. nov., *Neobacillus* gen. nov., *Metabacillus* gen. nov., and *Peribacillus* gen. nov. [23]. Moreover, as a result of the creation of these new genera, 103 erroneously assigned species, that were insufficiently related to the genus *Bacillus*, were assigned to the new genera. The results above constitute an important step in elucidating the taxonomy of the *Bacillus* spp. and related genera. However, as indicated above, comprehensive studies are still needed for the correct classification of *Bacillus* spp. and related species.

## 3. Occurrence of Spore-Forming Bacteria in Arable Soils

Bacteria of the genus *Bacillus* and related genera are widely distributed in the environment, e.g., in soil, air, water, animals, plants, or sediments [50–53]. This group of bacteria plays a particularly important role in the soil, including the decay of matter [54], promotion of plant growth, and protection against phytopathogens.

A very good and widely used tool for assessing the abundance of bacteria is the next-generation sequencing (NGS), including 16S rRNA genes sequencing. However, due to the still existing limitations of sequencing technologies, most studies present the abundance of bacteria at high taxonomic levels, i.e., phyla or orders, and rarely present the abundance of bacteria at the genus level, which is a subject to much greater error [55,56].

The phylum Firmicutes is one of the dominant phyla in cultivation soils. Its relative abundance in the soil ranges from 2% to about 20% depending on agrotechnical practices used, including crop rotation systems and fertilization type [57–61]. In general, the Firmicutes type is more abundant in soils from crop rotation than in soils from continuous cropping [62–64]. The reason for these patterns is probably a greater influence of crop residues and decomposing roots in the soil from crop rotation compared to monoculture soil. For instance, in a greenhouse experiment, Li et al. [59] detected a higher number of sequences assigned to the phylum Firmicutes in soil (Mollisol with sandy loam texture) derived from rotation (tomato/potato-onion) compared to monoculture (tomato). The same patterns were noted for the genus *Bacillus*. The abundance values obtained by the authors at the level of the phylum Firmicutes and the genus *Bacillus* did not exceed 10%. However, there are also cases where more Firmicutes are detected in monocultures than in rotations, or in longer monocultures than shorter ones. For example, in the soil from the Morrow Plots experiment (USA), the relative abundance of the phylum Firmicutes ranged from a few to a maximum of 14%. Its abundance was dependent on soil management; in this case, the highest value was recorded in soil from a maize monoculture, while the lowest abundance of sequences assigned to the phylum Firmicutes was noted in soil from a maize-soybean rotation [6]. Similarly, Zhao et al. [60] observed a significantly increased number of sequences belonging to phylum Firmicutes in soil from 15- and 22-year continuous cropping of cucumber in comparison with cucumber grown for only one year. Earlier, Zhao et al. [65] noted similar patterns in continuous cropping of coffee. However, these authors did not find specific reasons for this phenomenon [60,65]. Hence, further studies are needed to find parameters that have a considerable role in shaping the abundance of the phylum Firmicutes including *Bacillus* spp. and related genera, e.g., identifying detailed correlations between physical and chemical properties of the soil and the abundance of bacteria belonging to the phylum Firmicutes in differently managed soils. For example, Alami et al. [66] observed robust correlations between the phylum Firmicutes and the physicochemical properties of arable soil including continuous cropping of maize and cabbage continuous cropping of cabbage (Hubei province, China); total phosphorus, avail-

able potassium, and available boron contents were positively correlated with the phylum Firmicutes. Furthermore, a study on the effect of continuous cotton cultivation (20 years) on the bacterial communities of the soil showed a positive correlation between the number of the OTUs of the phylum Firmicutes and the EC of the soil [66].

In addition, fertilization also affects the abundance of the phylum Firmicutes in the soil. Particularly because most members of the phylum Firmicutes are considered copiotrophs which are fast-growing microorganisms that prefer environments rich in C and N [67]. For instance, Li et al. [68] also found a several percent abundance of the phylum Firmicutes in fertilized soil (rice-rape rotation), and the highest number of OTUs belonging to the phylum was found in soil fertilized with NPKS (NPK + straw). Similar values were also found by Zeng et al. [58] who observed an abundance of the phylum Firmicutes at an average of 7% (the highest value was 10%) in soil fertilized with nitrogen fertilizer. Dang et al. [69] observed a significant increase in the abundance of Firmicutes in soil fertilized with manure (compared to the controls) across the globe, and detected a positive correlation between the SOC content and the abundance of the phylum. Furthermore, Francioli et al. [70] noted more OTUs assigned to the phylum Firmicutes in farmyard manure (FYM) fertilized soils compared to mineral fertilization (in a long-term fertilization trial). Hartmann et al. [71] also observed higher abundance of the phylum Firmicutes in long-term FYM fertilization in comparison with mineral fertilization. Similar findings were noted in a study on the effects of various treatments on the microbial community of bulk and rhizosphere soil [72]. Importantly, it was also found that manure fertilization is a factor influencing the bacterial community (including the abundance of Firmicutes) more strongly than the method of cultivation, including monoculture and crop rotation [31,73].

In conclusion, it should be noted that the abundance of bacteria of the phylum Firmicutes in soil may also be influenced by other agronomic treatments such as the use of plant protection agents. Thus, the study results may also have been caused by the heterogeneity of agricultural practices, as previously recorded by Soman et al. [6]. Moreover, the discrepancies in studies in this aspect may be an effect of the diversity of soils around the world, e.g., in terms of physical properties.

## 4. Cellulolytic Properties of *Bacillus* Spp. and Related Genera

### 4.1. Cellulases

Cellulose is the most common (bio)polymer on earth, made of glucose linked by β-1,4-glycosidic bonds. It contains two types of regions—crystalline and amorphous regions [74]. Hence, an important group of microbes that are participating in the element's circulation in the soil are microorganisms that decompose cellulose [75]. Soil properties such as pH, organic carbon content, nitrogen content, and moisture impact microbial cellulose degradation. The process of cellulose degradation depends on the presence of a complex of enzymes belonging to the class of O-glycoside hydrolases, including the three main cellulases [74]. Cellulolytic enzymes include: (i) endo-β-1,4-glucanases (EC 3.2.1.4) whose mechanism of action is based on random degradation of β-1,4-glycosidic bonds in amorphous regions of cellulose–endoglucanase activity is measured using cellulose derivatives, for instance, semi-soluble carboxymethylcellulose (CMC); the enzyme that degrades CMC is carboxymethylcellulase (CMCase); (ii) exo-1,4-β-glucanases (EC 3.2.1.91) that separate single molecules of glucose and cellobiose from reducing or non-reducing ends of the cellulose. Exoglucanases include e.g., avicelase–microcrystalline–cellulose (Avicel) degrading enzyme; and (iii) β-glucosidase whose mechanism of action is the conversion of cellobiose into glucose (EC 3.2.1.21) [76,77]. The synergistic cooperation of the above-mentioned enzymes and, in particular, the presence of a processive exoglucanase is required for cellulose degradation [Figure 2].

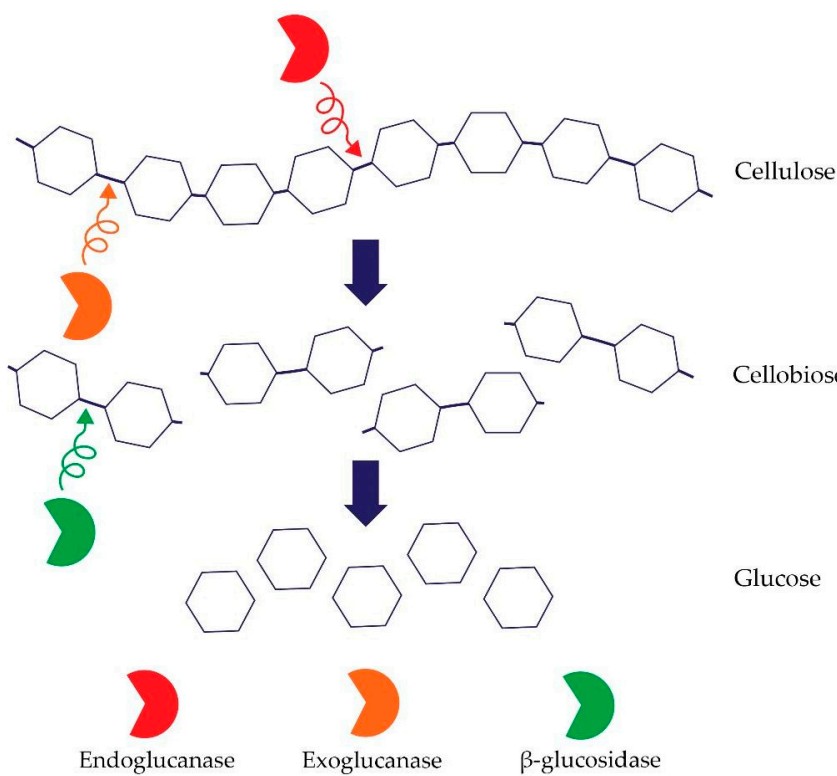

**Figure 2.** Mechanism of action of cellulases (own figure).

Cellulases have a modular structure that contains catalytic modules (CM) that act synergistically and/or with non-catalytic modules, i.e., substrate-binding modules [78]. In the case of cellulases, substrate-binding modules are called cellulose–binding modules (also called carbohydrate-binding module) (CBMs) and they can have affinity for amorphous or crystalline cellulose as well as binding to other similar polymers composed of carbon chains [79]. Due to the enzyme's ability to bind to cellulase, the local concentration of the enzyme increases, resulting in better substrate degradation efficiency. Some CBMs also have a structural function of stabilizing the catalytic module or altering its activity, for instance by inserting a substrate molecule into a substrate pocket [79]. Although the binding of cellulase by CBM is very stable, the enzyme can still diffuse across the substrate surface and, in some cases, CBM can also catalyze the breaking of non-covalent bonds located between the cellulose chains of crystalline cellulose [80].

*4.2. Structural Diversity of Cellulases*

Cellulases belong to the glycoside hydrolases (GH). The classification of GH is based on similarities in amino acid sequence and is included in the carbohydrate-active enzymes (CAZy) database. The CAZy database contains the CAZy families and subfamilies and is very dynamically updated. Due to the large differences in amino acid sequences, the GH group is remarkably heterogeneous and is divided into as many as 165 families [81]. The enzymes involved in cellulose degradation are classified in the following families of glycoside hydrolases: endoglucanases in families 5, 6, 7, 8, 12, 44, 45, 48, 51, 64, 71, 74, 81, 87, 124, 128, exoglucanases 5, 6, 7 and 4, and β-glucosidases in families 1, 3, 4, 17, 30, and 116 [82,83].

Sequence diversity can be related to a distinct modular architecture. It has been shown that the domain architecture in fungi is not very complex. However, in bacterial cellulases, there are many combinations of domain architectures, even though most sequences consist of a single catalytic domain [84]. In terms of carbohydrate-binding modules, CBM2 is related to the cellulolytic GH families and is found in the following families–GH5, GH12, GH44, GH45, GH48, GH51, and GH74. So far, it has been shown that the common CBM2

domain (in bacterial cellulases) in most cases binds cellulose, and less often chitin and xylan. Importantly, CBM2 is often found together with other accessory domains including CBM3 and CBM4, as well as catalytic domains [85]. In the terms of bacteria of the genus *Bacillus*, *B. licheniformis* possesses the H1AD14 gene encoding an endoglucanase belonging to the GH9 family and the cellulase has a CBM3 domain that is attached to the C-terminal end and plays a significant role in substrate degradation [86]. CBM3 has also been detected in a cellulase belonging to the GH9 family in *B. pumilus* [87]. Interestingly, Honda et al. [88] found that a unique chitinase domain in *B. thuringiensis* enabled binding to both crystalline chitin and cellulose, indicating that CBMs with affinity to multiple substrates could contribute to the increased occurrence of multifunctional hydrolytic enzymes [88]. On the other hand, previously, in the *B. subtilis* IFO 3034, an endoglucanase was detected that possessed a microcrystalline cellulose-binding domain but was unable to degrade microcrystalline cellulose [89]. Also in species related to *Bacillus* spp. CBMs were detected; in the *Paenibacillus lautus* BHU3 as many as four domains were detected, including CBM6, CBM46, CBM56, and CBM9, showing affinity for amorphous cellulose [90]. The CBM9 was also found in the genome of *P. dendritiformis* CRN1 [91].

Furthermore, cellulases belonging to different families have various protein fold structures, including the $(\beta/\alpha)8$ barrel fold, which is found in the GH5, GH44 and GH51 families, modified $\alpha/\beta$ barrel in family GH6, $\beta$-jelly roll—GH7 and GH12, the 7-fold $\beta$-propeller (GH74), $(\alpha/\alpha)6$ barrel—GH8, GH9 and GH48, the superhelical fold—GH124, and modified $\beta$ barrel (GH45) [81]. Importantly, within a single GH family, structures are globally conserved, but sequences can be remarkably different. For example, GH5, one of the largest GH families, is currently divided into 166 subfamilies on the basis of sequence similarity, with only eight residues conserved across the family, including two catalytic glutamic acid residues [92]. In conclusion, GH families exhibiting different classes of protein folds have evolved to bind and degrade the same substrate, indicating that cellulolytic enzymes may have evolved independently and may be derived from many evolutionary origins, but have converged functionally [81]. Similar patterns regarding the evolution of cellulases can be inferred from the large number of cellulose-binding domains.

*4.3. Cellulases Genes*

Referring to the number of cellulases, it can be concluded that cellulolytic enzymes are highly diverse, which is further manifested in the large number of genes that are responsible for encoding these hydrolases. The number of genes encoding cellulases exceeds 100 [83,93]. As mentioned earlier, the reason for such a large number of cellulase-encoding genes may be due to independent evolution [81]. In fungi, the genes encoding cellulases in bacteria are located on a chromosome [94]. The spatial organization of these genes may differ between microorganism species, for example, in the bacterial species *Clostridium thermocellum* there is a random distribution, whereas in *C. cellulovorans* "clustered" distribution in a cluster occurs [95,96]. The cellulosome gene cluster in *C. cellulovorans* is about 22 kpz in size and contains nine genes encoding cellulosome domains with a putative transposon gene in the flanking region. A similar organization was also detected in the chromosome of the bacterial species *C. acetobutylicum* and *C. cellulolyticum*, suggesting the presence of a common bacterial ancestor of the clostridia [97]. In contrast, in fungi the genes encoding cellulases are usually distributed randomly, in which case each gene has its own transcriptional regulation. Only in exceptional cases, e.g., in *Phanerochaete chrysosporium* (fungus), the cellulase genes form a three-gene cluster [94].

In terms of *Bacillus* spp., in the genome of *B. licheniformis* [98] detected two clusters of genes involved in the cellulose decomposition. For instance, in the genome of the strain *B. subtilis* 168 no equivalents of the cluster were found. The enzymes encoded by the first gene cluster are likely endoglucanases belonging to the GH9 and GH5 families, and the probable cellulase–1,4-β-cellobiosidase belonging to GH48 and the potential β-mannanase belonging to GH5. Importantly, β-mannanase (GH5) and endoglucanase (GH9) contain carbohydrate-binding modules. In addition, with the exception of 1,4-β-cellobiosidase

belonging to GH48, all gene proteins encoded have secretory signal peptides and all have homologs with *Bacillus* spp. but other than *B. subtilis* [98]. Researchers also detected a second cluster—encoding a probable β-glucosidase (from the family GH1). In addition, a second β-glucosidase gene (from the family GH3) was found at an unrelated locus in the genome. Importantly, the presence of these genes in the *B. licheniformis* genome indicate the possibility of complete degradation of cellulose [98]. Furthermore, 4 genes responsible for encoding β-glucosidase and 1 gene encoding endoglucanase were noted in strain *B. amyloliquefaciens* TL106. The β-glucosidases encoded by these genes belong to the GH1 and GH73 families, and the endoglucanase belongs to the GH5 family [99]. Moreover, Carbonaro et al. [100] analyzed the genome of *Alicyclobacillus mali* FL18 to find new cellulose–degrading enzymes. The analysis revealed four genes belonging to the GH1, GH9, GH51, and GH94 families, of which GH1 and GH94 legitimately hydrolyse short oligosaccharides, and a gene from GH1 encodes a β-glucosidase. In addition, the *A. mali* FL-18 genome also contained genes encoding two probable arabinofuranosidases, which belong to GH51. Interestingly, *A. acidocaldarius*, which is a close relative of the aforementioned species, also possesses two endoglucanases–CelA belonging to the GH9 family and CelB from the GH51 family [101,102]. At the same time, other authors have detected a large number of genes encoding various GH enzymes in the *P. polymyxa* genome, including cellulases belonging to GH 1, 3, and 5 [103].

Finally, it is worth adding that most of the cellulases described in metagenomic studies (different environments) have less than 70% homology with known cellulolytic enzymes, and some of them have no significant similarity to other glycosyl hydrolases, indicating that large numbers of new cellulolytic enzymes are still being found [104]. Moreover, approximately 40% of sequenced bacterial genomes contain at least one cellulase gene, but only 4% of these bacteria are known as true cellulase bacteria due to low cellulase diversity or a lack of gene expression [83].

### 4.4. Cellulosomes

Some bacteria exhibiting cellulolytic activity are capable of synthesizing and secreting enzyme multicomplexes called cellulosomes; the secreted proteins outside the bacterial cell take the form of spherical structures. Sometimes, individual cellulosomes are joined together to form so-called polycellulosomes. A single complex may contain up to a dozen proteins with different activities, including endoglucanase, cellobiase or hemicellulase, and lichenase [105,106]. Interestingly, most of the research on cellulosomes concerns the cellulosomes of the phylum Firmicutes [83]. First studies on cellulosomes were carried out on the anaerobic *Clostridium thermocellum*. As shown, *C. thermocellum* is capable of synthesizing an enzyme complex of more than 2000 kDa, which consists of fourteen different proteins with molecular weights ranging from 45 kDa to 210 kDa [107]. For instance, 15 genes encoding the presence of endoglucanases, two genes responsible for the expression of xylanases, two genes encoding cellobiase, and one gene encoding lichenase were detected in *C. thermocellum* strain NC1B 10682 [108]. Cellulosomes have also been detected in *Bacillus* spp. and related genera. However, to date, little research has been conducted on these genera. For instance, *B. megaterium* was found to be capable of producing a cellulosome (celluloxylanosomes) exhibiting avicelase, CMCase, and xylanase activity. In addition, van Dyk et al. [105] noted that the *B. licheniformis* SVD1 strain was capable of synthesizing multi-enzyme complex (MEC) with hemi-cellulolytic activity. The total molecular mass of the complex was about 2000 kDa. The enzymes included in the MEC hydrolyzed such compounds as xylan, mannose, pectin, and carboxymethylcellulose. However, the MEC was not able to bind Avicel cellulose and, despite several similarities to the cellulosome, was ultimately not identified as such [105,109]. Waeonukul et al. [110] studied an enzymatic complex from *P. curdolanolyticus* B-6 (in culture on Avicel microcrystalline cellulose). A single cellulosome had the ability to hydrolyze the Avicel cellulose and insoluble xylan. The researchers noted that the complex included such enzymes as avicelase, CMCase, cellobiohydrolase, β-glucosidase, xylanase, α-L-arabinofuranosidase, and β-xylosidase.

The total mass of the multicomplex was about 1600 kDa. Importantly, the isolated cellulosome degraded lignocellulose efficiently. In terms of the genus *Paenibacillus*, using transmission microscopy, cellulosome production was detected in *P. polymyxa* strains EG2 and EG14 [111,112]. Besides, using scanning electron microscopy, protuberances were observed indicating cellulosome production on the cell surface in the thermophilic strain *Brevibacillus* sp. JXL [113].

### 4.5. Cellulase Activity

Cellulose decomposition starts when cellulase adsorbs to cellulose. Referring to previous subsections, it should be stated that bacteria of the genus *Bacillus* and related genera are capable of producing several types of cellulases including CMCase, FPase, or Avicelase. Different bacterial species have distinct activities of cellulolytic enzymes, and significant differences within the same species or strains may also occur due to discrepancies in culture conditions of the studies conducted on the topic. For instance, Acharya and Chaudhary [114] observed a CMCase activity of 0.300 U mL$^{-1}$ in *Bacillus licheniformis* MVS1 (medium with beef extract). While Shajahan et al. [115], using response surface methodology in *Bacillus licheniformis* NCIM 5556, recorded a CMCase activity of 42.99 U mL$^{-1}$ (medium contained CMC—19.21 g L$^{-1}$, CaCl$_2$—25.06 mg L$^{-1}$, Tween 20—2.96 mL L$^{-1}$, and temperature 43.35 °C).

The type of cellulose, medium composition, temperature and pH are most important for cellulase activity [2]. So far, depending on the strains, it has been found that the type of cellulose used as substrate induces cellulolytic activity to a different extent. For instance, Sadhu et al. [116] observed that carboxymethylcellulose better induced Avicelase and CMCase production by *Bacillus* sp. MTCC10046 compared to other substrates including sucrose, starch, glucose, or maltose. Also, Akaracharanya et al. [117] recorded higher cellulase activity of *Bacillus* sp. P3–1 and P4–6 in culture based on CMC medium, compared to culture with cellulose powder-containing medium. Similar patterns were also reported by Thomas et al. [118] who observed that CMCase activity by *Bacillus* sp. SV1 was higher in the CMC medium, compared to Avicel cellulose-containing medium and other carbon sources, including mannitol, glycerol, lactose, or chitin. In addition, CMCase and Avicelase activities were also obtained by Dobrzynski et al. [2] who noted the highest activity of the two enzymes in the cultures of *Bacillus* sp. 8E1A with CMC. However, in the case of FPase (cellulose saccharifying enzyme), the highest activity value was recorded for the culture of the studied strain with Avicel cellulose. Mihajlovski et al. [119] also reported slightly higher FPase activity in *P. chitinolyticus* CKS1 in a medium supplemented with Avicel compared to cultures with CMC. Similarly, in the case of thermophilic *Bacillus* sp. K-12, Kim and Kim [120] noted that FPase activity was higher when the strain studied by the authors was cultured in Avicel microcrystalline cellulose medium compared to other carbon sources. Interestingly, in contrast to previously cited reports, the strain *Bacillus* sp. K-12 also had high CMCase and Avicelase activity in cultures with Avicel cellulose. It is worth mentioning that the differences between studies may result from a number of factors including culture conditions.

Another important factor that affects the activity of cellulases produced by *Bacillus* spp. and related genera is temperature. According to the studies cited below, the optimum temperature range for cellulase activity ranges from 20 °C to 80 °C, depending on the strain and type of enzyme. For instance, Kazeem et al. [121] observed that a temperature of 20 °C is optimal for the production of FPases in the strain *B. licheniformis* 2D55. Cellulases produced by *B. pseudomycoides* (grown on sugarcane bagasse medium) have a slightly higher optimal temperature −40 °C (within 72 h of incubation) [122]. Interestingly, Li et al. [123] detected optimal cellulase activity in the thermophilic strain at 50 °C, and below this value the activity of enzymes significantly decreased. On the other hand, optimum temperature values for cellulase activity exceeding 70 °C have been recorded for activity of CMCase and Avicelase produced by *Geobacillus thermoleovorans* T4 (70 °C) and CMCase produced by *Bacillus* sp. DUSELR13 (75 °C) [124,125]. Similar patterns for *Bacillus* sp. 8E1A were

observed by Dobrzyński et al. [2]. Importantly, thermophilic cellulases can potentially be used in various industries including textile, biofuel, and agriculture [2].

In terms of the optimal pH for cellulase activity, the range of values is as wide as for temperature; according to current reports, the highest activity of cellulases produced by *Bacillus* spp. and related genera is recorded in the pH range from 3 to 10. For example, Mihajlovski et al. [119] observed that the avicelase produced by the strain was most active at about pH 5. Similar results were reported by Seo et al. [126] whose *B. licheniformis* strain produced cellulases with high activity in the pH range of 4.0–6.0. While, in a study by Dobrzynski et al. [2], the highest CMCase and Avicelase activities were noted at pH 7.0 and FPase at 6.0. Interestingly, the highest cellulase activities produced by the bacteria of the genus *Bacillus* were also detected at pH 9.0 [127]. Previously, similar patterns were also obtained, as shown in Table 2.

**Table 2.** Optimum temperature and pH for celullolytic activity.

| Strains | Egzoenzymes | Temperature Optimum | pH Optimum | References |
|---|---|---|---|---|
| *Anoxybacillus* sp. 527 | Avicelase | 70 °C | 6.0 | [113] |
| *Anoxybacillus flavithermus* EHP2 | CMCase | 75 °C | 7.5 | [26] |
| *Bacillus* sp. *K1* | CMCase | 50 °C | 6.0 | [128] |
| *Bacillus* sp. KSM 330 | CMCase Avicelase | 45 °C | 5.2 | [129] |
| *Bacillus* sp. No.1139 | CMCase | 50 °C | 9.0 | [130] |
| *B. licheniformis* | CMCase | 65 °C | 6.0 | [131] |
| *B. subtilis* YJ1 | CMCase Avicelase | 50–60 °C | 6.0 | [132] |
| *Paenibacillus* sp. *B39* | CMCase | 60 °C | 6.5 | [133] |
| *Paenibacillus terrae* ME27–1 | CMCase | 50 °C | 5.5 | [134] |

Importantly, the differences between the optimal conditions for the activity of cellulolytic enzymes result from the large variety of cellulases produced by the spore-forming bacteria of the genus *Bacillus* and related genera.

Moreover, the activity of cellulases is also affected by other parameters of the media or solutions. Gaur and Tiwari [135] found that the cellulase activity of *B. vallismortis* RG-07 was stimulated by Tween-60, $Ca^{2+}$, mercaptoethanol, and NaClO. While the cellulase activity of *Lysinibacillus xylanilyticus* was stimulated by the presence of $CaCl_2$ nanoparticles in medium [136].

Importantly, some of the spore-forming strains of cellulolytic bacteria are already being used to convert lignocellulosic waste. For instance, the activity of *P. polymyxa* ND24 was studied in a 5-L laboratory bioreactor where the cellulosic substrate in the medium was sugarcane bagasse; the strain showed the highest endoglucanase activity after 72 h of incubation. The sugarcane hydrolysate was then used for biogas production; the authors suggest that the obtained results support the use of *P. polymyxa* ND24 for cost-effective bioprocessing of lignocellulosic biomass [137]. In turn, other authors have used strains from the genus *Bacillus* to treat rice straw in order to increase the biomethane fermentation efficiency. The study, using multiple strains, demonstrated that the use of mixtures of different bacterial strains was more effective than the use of single bacterial strains, due to an increase in the pool of cellulases present in the process. Finally, the authors concluded that the choice of a mixture of strains from the genus *Bacillus*, which decompose lignocelluloses, can be robust catalysts for the processing of biomass from these wastes [138].

However, despite such a large number of bacterial strains of the genus *Bacillus* and related ones that produce cellulases, there is still little research on the practical aspect of their use, including the utilization and conversion of lignocellulosic biomass. Nevertheless, potentially, cellulolytic bacteria of the genus *Bacillus* spp. and their cellulases can be used: (i) in the textile industry (for instance for biostoning of jeans); (ii) in biorefining; (iii) in biogas and biofuel production; (iv) in agriculture including biodegradation of lignocellulosic waste and biocontrol of fungal phytopathogens; (v) in the paper industry (coadditive in

pulp bleaching); (vi) in detergents (cellulose-based detergents); (vii) in the food industry including release of the antioxidants from fruit and vegetables, and improved texture and quality of bakery products; (viii) and for improving carotenoids extraction or improving olive oil extraction [139,140].

## 5. Promoting Plant Growth by the Bacteria of the Genus *Bacillus* and Related Genera

Bacteria of the genus *Bacillus* and related genera are also classified as plant growth-stimulating bacteria [53,141–143]. Bacteria from this group are capable of promoting plant growth either directly or indirectly. Mechanisms of direct promotion of plant growth include i.a. production of phytohormones including indole-3-acetic acid (IAA), cytokinins, and gibberellins, production of nitrogenase thanks to which bacteria fix atmospheric nitrogen (N) and make it available to plants, and the possibility of solubilizing phosphorus. Indirect mechanisms, on the other hand, include for instance production of antibiotics including cyclic lipopeptides, and enzymes degrading fungal cell walls [144–147].

So far, plant growth-promoting abilities have been detected in a very large number of bacteria belonging to the genus *Bacillus* or related genera. Bacteria from this group have promoted plant growth both under controlled and field conditions. Because of the greater value of studies under field conditions, several examples of such studies are presented in the review. For instance, inoculation of rice seedlings with *B. pumilus* TUAT-1 supplemented with N fertilizer led to an increase in height, biomass, and chlorophyll content of rice plants [148]. Besides, Ali et al. [149] showed that *B. cereus* (potassium solubilizing strain) increased the plant's height and shoots' dry weight. Importantly, compared to plants that were not inoculated, the application of the strain resulted in an increase of about 20% in potato yield. Moreover, the application of *Paenibacillus triticisoli* BJ-18 led to an increase in N, P, and organic matter contents in soil and enhanced nitrogenase activity and wheat yield [150]. Interestingly, in comparison to the control, the application with the strain also increased the biodiversity of rhizosphere bacterial communities and led to an increase in the abundance of the genus *Paenibacillus* in the inoculated soil, which also resulted in a high abundance of genes encoding nitrogenases. Furthermore, the inoculation with *P. triticisoli* BJ-18 also increased the abundance of native plant growth-stimulating bacteria of the genera *Bacillus* and *Podospora* [150].

Besides Okoroafor et al. [151], after applying *B. velezensis* FZB42 (formerly *B. amyloliquefaciens* FZB4) in maize and common sunflower cultivations, detected over 20% increase in biomass production in each of the crops. Moreover, inoculation with the tested preparation increased the bioavailability of soil elements. Interestingly, the study on winter wheat cultivation by Stepien et al. [152] is an example of a field experiment with *Bacillus* and related bacteria. The researchers demonstrated that the combination of mineral fertilization and three bacteria-*Paenibacillus azotofixans*, *B. megaterium*, and *B. subtilis*-significantly increased wheat grain yield compared to the application of mineral fertilization alone. In addition, the bacteria significantly increased the leaf greenness index SPAD at two time points, and together with NPK fertilization, significantly increased the content of two forms of nitrogen ($N-NO_3$ and $N-NH_4$) and phosphorus in the soil.

Another example of research using a bacterial consortium with *Bacillus* spp. is an experiment using *B. cereus* AR156, *B. subtilis* SM21, and *Serratia* sp. XY21 (BBS) strains applied to phytophthora-infested sweet pepper [153]. Compared to the control, the application of BBS reduced the occurrence of phytophthora blight and enhanced the fruit quality and soil properties. BBS also significantly increased the abundance of the bacterial genera *Burkholderia*, *Comamonas*, and *Ramlibacter*, which were negatively correlated with disease severity; moreover, the abundance of these genera were associated with organic carbon, ammonia nitrogen, potassium, and available phosphorus. These patterns suggest that changing the bacterial community improved the soil properties and reduced the phytopathogen development.

Importantly, there are still not enough studies in field conditions, especially those showing the effect of the inoculants used on the native microbiota whose biodiversity and

taxonomic composition have the greatest influence on the biochemical processes of the soil. Finally, field studies with a wide range of parameters will bring inoculants closer to commercialization. However, there are already quite a number of commercial preparations containing *Bacillus* and related bacteria, for example biofertilizers, biofungicides, or biopesticides (listed in Table 3).

**Table 3.** Commercial preparations containing *Bacillus* and related bacteria.

| Bacteria | Application | Mechanism | Commercial Biopreparation | Reference |
|---|---|---|---|---|
| *B. subtilis* C-3102 | biofertilizer | for example: IAA production | Thervelics® | [154] |
| *B. subtilis* | biofertilizer | phosphate solubilization | BCMF | [155] |
| *B. megaterium* (combination with *Azotobacter chroococcum*, *Azospirillum brasilense*) | biofertilizer | phosphate solubilization | Azoter® | [156] |
| *P. azotofixans*, *B. megaterium* and *B. subtilis* | biofertilizer | nitrogen fixation | no information available | [152] |
| *B. velezensis* D747 | biofungicide | cyclic lipopeptides | Double Nickel 55™ | [157] |
| *B. velezensis* FZB42 | biofungicide | antibiotic substances (polyketides and lipopeptides) | Taegro® | [157] |
| *B. velezensis* QST 713 | antifungal and antibacterial product | antibiotic substances | Serenade® ASO | [158] |
| *B. thuringiensis var. kurstakivar* | biopesticide | crystal proteins (Cry) production | BT-Biox WP® | [159] |
| *B. firmus* I-1582 | biopesticide | protection against nematode infection | VOTiVO® | [160,161] |

## 6. Conclusions

In summary, bacteria of the genus *Bacillus* and related genera constitute an important group of bacteria that populate soil and other environments in large numbers, but their taxonomy is still inadequately defined, due to, among other things, their great diversity and the selection of insufficiently suitable molecular and biochemical techniques to determine their relationship. Among this group of bacteria, cellulolytic bacteria are one of the most important, but knowledge about their occurrence in the soil environment is still limited, which is caused by methodological difficulties faced by scientists studying it. Most studies on the presence of cellulolytic bacteria in the soil are limited to determining the abundance of genes encoding cellulase, which, due to the diversity of these genes, makes it impossible to determine the abundance of individual groups of cellulolytic bacteria.

Moreover, despite dozens of isolates of *Bacillus* and related bacteria showing cellulolytic activity, still few of these bacterial strains are used, for example, to degrade lignocellulosic waste. Importantly, the amount of lignocellulosic waste generated by agriculture and other industries is steadily increasing, which, in an era of progressive agriculture and other industries generating large amounts of such waste, poses a huge environmental problem. Therefore, researchers should focus on studying the cellulolytic bacteria, e.g., in biogasification processes or other conversions, which could contribute to the commercialization of these bacteria.

**Author Contributions:** Conceptualization, J.D.; methodology, J.D.; software, B.W.; writing—original draft preparation, J.D. and E.B.G.; writing—review and editing, J.D., B.W. and E.B.G.; visualization, B.W.; supervision, J.D. and E.B.G.; project administration, J.D.; funding acquisition, B.W. All authors have read and agreed to the published version of the manuscript.

**Funding:** This research received no external funding.

**Institutional Review Board Statement:** Not applicable.

**Acknowledgments:** Many thanks to Katarzyna Rafalska for help in revising the English language and Aleksandra Wróbel for help in data visualization.

**Conflicts of Interest:** The authors declare no conflict of interest.

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
