# Peer review of "Taxonomy, Ecology, and Cellulolytic Properties of the Genus Bacillus and Related Genera"

_agriculture, doi:10.3390/agriculture13101979_

Round 1

Reviewer 1 Report

Dear Authors,

Please find my comments below for further improvement of the manuscript:

1. Please discuss the old and new methods used to study taxonomy and how modern techniques are becoming helpful in identification and proper characterization of the microbes.

2. Also provide data regarding genes responsible for cellulase production, across different genera, the similarities and differences between them.

3. Gene sequence alignments to show sequence divergence , conserved regions and etc.

4. I believe that data provided is not enough, there can be made many subdivisions with in the main headings to make a review article comprehensiv and complete.

5. At the moment it is providing information but not enough to be considered for the review article.

Author should seriously think about incorporating more information with suitable structuring of the article.

Satisfactory

Author Response

Dear Reviewer 1

Thank you for your time and effort in reviewing our manuscript. The feedback has been invaluable in improving the content of the paper.

We have revised our manuscript according to all comments; the point-by-point responses below.

  1. Please discuss the old and new methods used to study taxonomy and how modern techniques are becoming helpful in identification and proper characterization of the microbes.

Thank you for your valuable feedback, in our paper we included examples of older techniques to identify bacteria and pointed out that the moment of development of molecular techniques was a turning point in the development of phylogenetics.

  1. Also provide data regarding genes responsible for cellulase production, across different genera, the similarities and differences between them.

Thank you for your valuable feedback, data regarding genes responsible for cellulase production had been provided in the text of manuscript.

  1. Gene sequence alignments to show sequence divergence, conserved regions and etc.

Thank you for your valuable feedback. We have introduced additional information about cellulase-encoding genes in the genomes of Bacillus and related bacteria. Unfortunately, to our knowledge there are no studies comparing the sequences of individual cellulase-encoding genes - which may be due to their vast number and diversity, which in turn may be influenced by their independent evolution (as highlighted in the text). However, we found, for example, a comparison of B. licheniformis cellulase to B. subtilis.

Moreover, most of the cellulases described in metagenomic studies (different environments) have less than 70% homology with known cellulolytic enzymes, and some of them have no significant similarity to other glycosyl hydrolases, indicating that large numbers of new cellulolytic enzymes are still being found. Therefore, systematizing the knowledge about genes encoding cellulase would require a lot of work of many bioinformaticians.

Nevertheless, in order to emphasize the great diversity of cellulases, we added a text about the diversity at the level of protein structures including domains and protein fold structures.

  1. I believe that data provided is not enough, there can be made many subdivisions with in the main headings to make a review article comprehensiv and complete.

Thanks for your feedback, we added new subsections: "Structural diversity of cellulases" and "Cellulases genes".

Reviewer 2 Report

The review article is good overall and provides insights into the biochemical and industrial usage of bacteria. The English language is pretty clear to the reader but the structure of the paragraphs isn't arranged and wrapped up well. The article doesn't have enough illustrations and figures. 

I recommend doing a phylogeny to see the variation among CX enzymes of different bacteria. Please spot the functional domains within those enzymes. 

I also suggest you compare the efficiency of bacterial CX to the basidiomycetes fungi and see which is more efficient. 

Those suggestions will make the review article more sounding. 

The English language isn't perfect but it is okay and can be understood. Paragraphs need to be restructured in a scientific way. 

Author Response

Dear Reviewer 2

Thank you for your time and effort in reviewing our manuscript. The feedback has been invaluable in improving the content of the paper.

We have revised our manuscript according to all comments; the point-by-point responses below.

1.The English language is pretty clear to the reader but the structure of the paragraphs isn't arranged and wrapped up well. The article doesn't have enough illustrations and figures.

Thank you very much for your valuable feedback. We corrected the paragraphs and added an additional figure.

2. I recommend doing a phylogeny to see the variation among CX enzymes of different bacteria. Please spot the functional domains within those enzymes.

Thank you for your valuable feedback. We would like to state, that most of the cellulases described in metagenomic studies (different environments) have less than 70% homology with known cellulolytic enzymes and some of them have no significant similarity to other glycosyl hydrolases, indicating that a large number of new cellulolytic enzymes are still being discovered - this was also included in the text. At the same time, drawing attention to the difficulty of systematizing data on cellulases due to their great diversity being the result of probable independent evolution.Therefore, systematizing the knowledge of the genes encoding cellulases (and aa sequences) will require the long-term work of many bioinformaticians.

Nevertheless, we have introduced more information about cellulase domains to show how diverse they are; we have given examples of the occurrence of domains in various Bacillus and related bacteria. In addition, we also showed the diversity of cellulases in terms of protein fold structures. And finally, we expanded on cellulase genes into a new subsection.

3. I also suggest you compare the efficiency of bacterial CX to the basidiomycetes fungi and see which is more efficient.

The comparison of cellulase-producing microorganisms is a very difficult issue due to the discrepancies occurring in the studies, including differences in fermentation conditions, composition of the medium and type of substrate - cellulose. Moreover, direct comparison of cellulolytic activity in different reports is often impossible due to differences in assay methods and activity units used - despite the fact that there is an IUPAC approved methodology for the determination of cellulolytic activity. Nevertheless, we have included information in the text that class Basidiomyctes are probably better producers of cellulases, but they are difficult to compare with bacteria due to the above facts.

4. Responses to the other suggestions in the attachment.

Thank you for all your suggestions, we have made the changes in review mode.

- we added the purpose of the review

- we corrected all single words and sentences

- we added examples of PGPB

Round 2

Reviewer 1 Report

There is always room for improvement, however changes made were according to the comments.

Satisfactory

Author Response

Dear Reviewer,

Thank you for your time in reviewing our manuscript.

Authors

Reviewer 2 Report

The review has an interesting topic that sheds light on the value of Bacillus spp. in cellulose degradation. The review has touched on the basic knowledge of the secreting bacteria and their enzymatic products. However, more information is required to add value to the review. 

First, the aim of this review (line 64) is dull and unclear. It requires more improvement. Please be specific and increase your aims. 

2- I would discuss the variations, if any, between bacterial strain capacity in cellulose degradation.

Does all Bacillus spp. produce the same enzymes, same features, same amounts?

Please give at least 10 examples of how this is valuable to the industry.

The last couple of sentences of the abstract should summarize the aim of this review. 

In Figure 1, Please choose a photo with a better resolution. Also, please aim for the spores and bacterial cells (Please be aware that bacterial cells do not produce ectospores, they always produce endospores, and thus they require a certain type of acid stain to be visualized). 

In Figure legends, please add adequate descriptions for the information required to understand the photo. The current legend is not acceptable. 

Please add schematic diagrams to summarize your explanations (Line 410), and anywhere else when possible. 

Please be aware that this topic is more related to environmental microbiology and will more likely be welcomed and appreciated in industrial and ecological microbiology-related journals. 

Author Response

Dear Reviewer

Thank you for your time in reviewing our manuscript. The subsequent feedback was also very useful. 

We have revised our manuscript according to all comments; the point-by-point responses below.

  1. First, the aim of this review (line 64) is dull and unclear. It requires more improvement. Please be specific and increase your aims. 

Thank you for your valuable feedback. We have modified our aim according to your suggestion. 

  1. I would discuss the variations, if any, between bacterial strain capacity in cellulose degradation. Does all Bacillus spp. produce the same enzymes, same features, same amounts?

Thank you for your valuable feedback. We have introduced some new sentences. 

  1. Please give at least 10 examples of how this is valuable to the industry.

Thank you for your valuable feedback.  We have added more than 10 examples of the application of cellulases in industry.

  1. The last couple of sentences of the abstract should summarize the aim of this review. 

Thank you for your valuable feedback. We have introduced a few sentences about it.  

  1. In Figure 1, Please choose a photo with a better resolution. Also, please aim for the spores and bacterial cells (Please be aware that bacterial cells do not produce ectospores, they always produce endospores, and thus they require a certain type of acid stain to be visualized). 

In Figure legends, please add adequate descriptions for the information required to understand the photo. The current legend is not acceptable. 

Thank you for your valuable feedback. We have added a better resolution photo, highlighted objects and improved the photo description.

  1. Please add schematic diagrams to summarize your explanations (Line 410), and anywhere else when possible. 

Thank you for your valuable feedback. We have added a diagram.